# Five years of pharmaceutical industry funding of patient organisations in Sweden: Cross-sectional study of companies, patient organisations and drugs

Shai Mulinari[1]*, Andreas Vilhelmsson[1], Emily Rickard[2], Piotr Ozieranski[2]

1 Department of Sociology, Lund University, Lund, Sweden, 2 Department of Social and Policy Sciences, University of Bath, Claverton Down, Bath, England, United Kingdom

* shai.mulinari@soc.lu.se

**Data Availability Statement:** The dataset used in this study is available here: http://doi.org/10.5281/zenodo.3886331.

## Abstract

### Background

Many patient organisations collaborate with drug companies, resulting in concerns about commercial agendas influencing patient advocacy. We contribute to an international body of knowledge on patient organisation-industry relations by considering payments reported in the industry's centralised 'collaboration database' in Sweden. We also investigate possible commercial motives behind the funding by assessing its association with drug commercialisation.

### Methods

Our primary data source were 1,337 payment reports from 2014–2018. After extraction and coding, we analysed the data descriptively, calculating the number, value and distribution of payments for various units of analysis, e.g. individual companies, diseases and payment goals. The association between drug commercialisation and patient organisation funding was assessed by, first, the concordance between leading companies marketing drugs in specific diseases and their funding of corresponding patient organisations and, second, the correlation between new drugs in broader condition areas and payments to corresponding patient organisations.

### Results

46 companies reported paying €6,449.224 (median €2,411; IQR €1,024–4,569) to 77 patient organisations, but ten companies provided 67% of the funding. Small payments dominated, many of which covered costs of events organised by patient organisations. An association existed between drug commercialisation and industry funding. Companies supported patient organisations in diseases linked to their drug portfolios, with the top 3 condition areas in terms of funding–cancer; endocrine, nutritional and metabolic disorders; and infectious and parasitic disorders–accounting for 63% of new drugs and 56% of the funding.

**Funding:** This work was supported by grants from the Swedish Research Council for Health, Working Life and Welfare (www.forte.se) (FORTE #2016-00875 to SM (PI) and PO) and the Crafoord Foundation (www.crafoord.se) (to SM (PI) and PO and AV). The funders had no role in study design, data collection and analysis, decision to publish, or preparation of the manuscript.

**Competing interests:** The authors have declared that no competing interests exist.

## Conclusion

This study reveals close and widespread ties between patient organisations and drug companies. A relatively few number of companies dominated the funding landscape by supporting patient organisations in disease areas linked to their drug portfolios. This commercially motivated funding may contribute to inequalities in resource and influence between patient organisations. The association between drug commercialisation and industry funding is also worrying because of the therapeutic uncertainty of many new drugs. Our analysis benefited from the existence of a centralised database of payments–which should be adopted by other countries too–but databases should be downloadable in an analysable format to permit efficient and independent analysis.

## Background

Patient organisations are increasingly involved with healthcare services and systems [1, 2] including in relation to pharmaceuticals [3, 4]. The increasing scope of activity results in patient organisations encountering many challenges in developing their capacity and expertise [1, 5]. Some of these challenges may be addressed by funding and other support provided by pharmaceutical companies [6, 7, 8, 9, 10, 11]. However, partnering with companies also raises concerns about conflicts of interest weakening patient organisations' independence from commercial agendas [4, 11,12, 13, 14, 15, 16, 17, 18] that do not always coincide with the interest of patients and public health [19, 20]. For example, several industry-funded patient groups at the European Union level supported the industry's efforts to relax the ban on direct-to-consumer advertising, but this agenda was strongly opposed by patient groups that did not accept industry funding [21].

One response to these concerns has been to enhance the transparency of financial relationships between the two sides [22]. This is part of a global policy trend in which transparency, primarily understood as public disclosures, is applied to individuals' and organisations' ties to pharmaceutical companies [23, 24]. Perhaps the most comprehensive disclosure initiative pertaining to patient organisations has been introduced by the European Federation of Pharmaceutical Industries and Associations (EFPIA), representing pharmaceutical companies operating in Europe, via its Patient Organisation Code of Practice [25]. Implemented through industry self-regulation since 2012, the Code requires pharmaceutical industry trade group members to disclose payments to patient organisations, which are typically published as annual reports on company websites [26].

As a result of this and similar transparency initiatives elsewhere in the world, there has been a spur in research on pharmaceutical company funding of patient organisations in the United Kingdom (UK) [4, 22, 26], Australia [27] and United States [28] using industry disclosure data. Industry disclosures have clear advantages over other data sources, such as surveys of patient organisations [14, 29] or their websites [30, 31], due to their greater coverage and standardisation [26]. The emerging picture from this research is that many patient organisations receive industry funding but that relatively few receive a large share of it [26, 27]. Most of the funding goes to research and public engagement, including advocacy, campaigning and disease awareness, and lobbying [26]. Furthermore, companies predominantly fund organisations in commercially high-profile areas, such as cancer and diabetes, indicating mercantile rather than philanthropically motivated support [26]. However, only a few countries have been surveyed making generalisations problematic, especially since the pattern of patient organisations' relations with industry may vary between countries [28].

Here, we contribute to building an internationally comparative body of knowledge by considering industry reports of payments to patient organisations in Sweden, using the same methodology as in the previous UK study [26]. Sweden has a history of pharmaceutical industry self-regulation [32] and a large patient organisation community: roughly 1 in 20 Swedes are affiliated to a patient organisation [33]. Another reason for selecting Sweden for this case study is that the Swedish pharmaceutical industry trade group (Läkemedelsindustriföreningen, LIF) has gone further than trade groups in other European countries that deem it sufficient to disclose payments on the sponsoring company's webpage, and which in the UK was shown to make it near-impossible to establish the scope of industry involvement for any patient organisation [22, 26]. In contrast–and consistent with recommendations from the UK [22]–since 2005 LIF has a centralised and searchable (albeit not downloadable) database where companies upload reports for every payment. The aims of this study were therefore to assess:

- the pattern of industry engagement with patient organisations by looking at available payment reports

- the association between drug commercialisation and industry funding of patient organisations

- the advantages of a centralised database for identifying and understanding the nature of industry funding of patient organisations

## Methods

### Data and coding

We extracted 1,412 payment reports from the Swedish pharmaceutical industry's trade group's "collaboration database" [34], covering a 5-year period, from January 2014 to December 2018. We selected a 5-year period to have a fairly large but still manageable sample, but also because industry rules allow reports to be deleted after three years which limited data availability. Data was collected in two phases: data from January 2014 to December 2016 were collected in November and December 2016, and data from January 2017 to December 2018 were collected in April and May 2019. Two authors (SM and AV) manually extracted the data into an Excel database, including the report title, project start and end dates, sponsoring company, recipient organisation, and project and payment descriptions, which are all contained in separate fields. SM extracted the data from January 2014 to December 2016 and AV the data from January 2017 to December 2018 in dialogue with SM. AV checked the data extracted by SM from June 2016 to December 2016 but found no discrepancies. Each entry was coded separately for payment category (e.g. sponsorship, grant, fees for service and consultancy) and goal (e.g. policy engagement, education and training) [26]. Given our focus on the national level, we aggregated local and regional patient organisations into their parent national association. Each patient organisation was coded according to their condition and disease area using ICD-10 (International Classification of Diseases, 10th revision). Most payments were reported in Sweden's currency, SEK. We expressed all payments in 2018 EUR using average annual exchange rates available from Sweden's Central Bank. To adjust for inflation, we used a yearly consumer price index obtained from Statistics Sweden (2014 = 4.76%, 2015 = 4.81%, 2016 = 3.79%, 2017 = 1.95%, 2018 = 0).

### Descriptive data analysis

To examine the pattern of the disclosed payments, we performed descriptive analyses calculating the number, value and distribution (median, interquartile range (IQR)) of payments. The

descriptive analysis was specified beforehand to the following units of analysis validated in a previous UK study [26]: all companies and patient organisations, individual companies and patient organisations, condition and disease areas, and payment categories and goals (S1 Table). In short, the payment category codes were initially devised based on the codes used by EFPIA to categorise pharmaceutical company payments to healthcare organisations (e.g. hospitals, universities, medical associations), specifically, "grants", "contributions to costs of events", "travel, accommodation and registration fees", "fees for service and consultancy", and "sponsorship"'. These codes were then supplemented with an inductive approach for any emerging payment categories that were unique to the patient organisation payment descriptions, such as "support and help" [22, 26]. Separately, payment goal was coded based on close iterative reading of payment descriptions and aggregating similar descriptions under the same codes [22, 26]. When coding payment goals we looked for the main purpose of activities funded by drug companies. Furthermore, for each patient organisation, condition and disease area, we identified the supporting companies as well as the main donor's share of the overall funding.

During the coding, we realised that a substantial amount of the funding–especially of policy engagement and advocacy–concentrated to the so-called Politician's Week held every summer, known nationally also as the Almedalen Week, which we therefore included as another unit of analysis. The Politician's Week can be described as a micro-cosmos of Swedish politics where politicians, ministerial advisors, governmental and special interest organisations, companies, think thanks, lobbyist, policy- and PR consultants and journalists, gather for debate, advocacy and lobbying, marketing, and social and political networking. Thus, Politician's Week funding may provide special insight into joint lobbying efforts.

## Association between drug commercialisation and industry funding

We began by assessing the concordance (Cohen's κ) between companies marketing drugs in specific ICD-10 disease areas and their funding of corresponding patient organisations. To make the analysis practical, we considered the top ten donors overall in the ten most funded disease areas. The online Swedish Medicines Compendium–FASS–has webpages for every pharmaceutical company which lists all products marketed in Sweden with weblinks to the Summary of Products Characteristics that contain information on approved indications. We used information on approved indications to determine whether or not companies marketed at least one drug in each of the ten selected diseases. For a drug to be considered marketed in a disease it had be indicated for the disease, e.g. drugs approved in HIV/AIDS patients for combating non-HIV infections, such as fungal infections, were not HIV drugs. Similarly, analgesics indicated for cancer patients were not cancer drugs.

In a separate analysis we calculated the correlation between the commercialisation of new drugs in ICD-10 condition areas and payments to patient organisations in those condition areas by all LIF companies over the study period. We performed the analysis at the level of broader conditions (e.g. neoplasms), rather than narrower diseases (e.g. breast or prostate cancers), because the former is less sensitive to the fact that our drug sample for this analysis only included drugs approved 2014–2018 and that drugs' indications may broaden over time. Furthermore, an important share of the funding, especially to cancer patient organisations, could not be differentiated at the level of specific diseases. In a first step, we compiled data on all new prescription drugs (i.e. New Active Substances (NAS) but excluding vaccines) from LIF companies that were approved by the European Medicines Agency (EMA) between 2014–2018, including information on date of approval, drug company and approved indication, based on information on the EMA and FASS websites. Next, we coded the drugs according to ICD-10

categories. Finally, using Spearman's rank-order correlation, we assessed the correlation between the number of new drugs approved in 2014–2018 for various ICD-10 conditions as well as the number of LIF companies marketing these drugs per ICD-10 condition area and, separately, the number and value of payments in ICD-10 condition areas. Spearman's rank-order correlation, rather than Pearson correlation, was chosen because of the large number of cancer drugs and payments to cancer patient organisations. A p-value below 0.05 was considered significant. Prism 8.2.1 for Macintosh (GraphPad Software Inc.) was used all statistical analyses.

## Results

### Overview of industry payments

Over the 5-year period, 2014–2018, 46 pharmaceutical companies reported 1,412 relationships with the local, regional or national branches of 77 patient organisations. However, 75 reports (5.3%) lacked information on the value of the payment, in violation of industry rules, and were subsequently excluded. This left us with 1,337 reports totalling €6,449,224, including 56 (4.2%) reporting zero-value payments (i.e. relationships without financial support). As shown in Table 1, there was an increase of annual payments between 2014–2016, reaching €1,583,581 (n = 298) in 2016, followed by a drop in 2017 and in 2018 –to €1,091,558 (n = 257). The funding landscape was dominated by small payments with roughly one quarter being smaller than €1,000 (26.0%; n = 348), and three quarters (77.9%; n = 1041) below €5,000 (median €2,411; IQR €1,024–4,569). However, the ten largest payments constituted roughly 15% of the total value (€942,623).

### Donors and recipients

There was a high concentration of payments among companies. The top ten donors provided 67.9% (€4,379,604) of the funding (Table 2; S2 Table for full table). Pfizer was the major donor with 14.8% (€954,234) of the reported funding whereas AbbVie had the greatest number of payments (10.9%; n = 146). Pfizer also made the largest single payment, worth €353,179, to the Swedish Breast Cancer Association in 2016 to support the creation of a website for information dissemination and advocacy (S3 Table). Pfizer also made the second largest payment to the same patient organisation in 2018 worth €97,497 supporting the same project. Besides Pfizer, AbbVie was the only other company that had more than one payment in the top ten.

As shown in Table 3, there was also a high degree of concentration of payments among patient organisations. Of the 77 patient organisations that received funding, the top ten recipients amassed 62.4% (€4,026,410), and the top recipient, the Swedish Breast Cancer Association, alone received 10.3% (€663,774), from just five companies. The Network Against Cancer–an umbrella coalition of cancer patient and advocacy organisations and networks–had the greatest number of company funders (n = 16) but was only the fourth most funded organisation.

**Table 1. Reported drug company payments to patient organisations in Sweden (2014–18).**

|  | 2014 | 2015 | 2016 | 2017 | 2018 | All years |
|---|---|---|---|---|---|---|
| Value of payments (€) | 1 261 999 | 1 350 447 | 1 583 581 | 1 161 639 | 1 091 558 | 6 449 224 |
| Median (IQR) (€) | 2 879 (1 152 to 5 758) | 2 912 (1 055 to 6 676) | 2 192 (937 to 3 933) | 2 117 (1 058 to 4 234) | 2 214 (887 to 3 900) | 2 411 (1 024 to 4 569) |
| No. of payments | 244 | 271 | 298 | 267 | 257 | 1 337 |
| No. of drug companies | 32 | 33 | 37 | 33 | 29 | 46 |
| No. of patient organisations | 39 | 45 | 54 | 50 | 40 | 77 |

**Table 2. Top ten drug companies reporting payments to patient organisations in Sweden (2014–18).**

| Company | Value of payments. € (%)[1] | n (%)[2] |
|---|---|---|
| Pfizer | 954 234 (14.8) | 92 (6.9) |
| AbbVie | 731 902 (11.3) | 146 (10.9) |
| Sanofi | 447 812 (6.9) | 87 (6.5) |
| Roche | 404 049 (6.3) | 131 (9.8) |
| Novartis | 403 508 (6.3) | 95 (7.1) |
| GlaxoSmithKline | 383 346 (5.9) | 43 (3.2) |
| Janssen | 335 870 (5.2) | 85 (6.4) |
| Bayer | 260 900 (4.0) | 57 (4.3) |
| Celgene | 243 880 (3.8) | 64 (4.8) |
| Amgen | 214 103 (3.3) | 64 (4.8) |
| **Total** | **4 379 604 (67.9)** | **864 (64.6)** |

[1] Percent of total value of payments (€6,449,224)

[2] Percent of total number of payments (1,337)

## Category and goal of funding

The top funding priority was supporting patient organisations' engagement with outside audiences (Table 4). Thus, the most common payment *category* was "contribution to costs of events" organised by the recipient patient organisation or third parties, for example an external meeting or lecture, attracting 35.7% (€2,300,750). Other main payment categories were "support and help" (20.5%; €1,325,363), "partnership arrangements" (15.8%; €1,021,968) and "sponsorships" (12.9%; €831,430).

**Table 3. Ten most funded patient organisations by reporting drug companies in Sweden (2014–18).**

| Patient organisation | Value of payments, € (%)[1] | n (%)[2] | Number of supporting companies | Main donor, € (%)[3] |
|---|---|---|---|---|
| Breast Cancer Association | 663 774 (10.3) | 62 (4.6) | 5 | Pfizer, 492 074 (74.1) |
| Blood Cancer Association | 568 107 (8.8) | 131 (9.8) | 10 | Novartis, 157 338 (27.7) |
| Association for gastro-intestinal diseases | 469 576 (7.3) | 49 (3.7) | 11 | AbbVie, 141 150 (30.1)[4] |
| The Network Against Cancer[5] | 397 451 (5.7) | 117 (8.7) | 16 | Celgene, 48 310 (12.2) |
| Heart and Lung Association | 381 914 (5.9) | 67 (5.0) | 10 | Pfizer, 84 253 (22.1) |
| Rheumatism Association | 353 739 (5.5) | 55 (4.1) | 13 | AbbVie, 124 141 (35.1) |
| Prostate Cancer Federation | 338 684 (5.2) | 134 (10.0) | 8 | Astellas, 148 674 (43.9) |
| Hemophilia Society | 321 315 (5.0) | 65 (4.9) | 11 | Bayer, 90 404 (28.1) |
| Neuro | 280 757 (4.3) | 50 (3.7) | 9 | Biogen, 126 091 (44.9) |
| 1.6 Million Club[6] | 251 093 (3.9) | 32 (2.3) | 8 | Bayer, 63 394 (25.2) |
| **Total** | **4 026 410 (62.4)** | **762 (57.0)** | | |

[1] Percent of total value of payments (€6,449,224)

[2] Percent of total number of payments (1,337)

[3] Percent of total value of payments received by the patient organisation

[4] In addition, AbbVie made two payments worth €62,677 together with Bristol-Myers-Squibb. Because no information was provided on how co-funding was shared between the two companies, the two payments are not included here.

[5] The Network Against Cancer is a coalition of cancer patient organisations, advocacy organisations and networks, such as Swedish Childhood Cancer Fund, Blood Cancer Association, Cancer Society of PALEMA (pancreas, liver, stomach and oesophagus), Swedish Association of Lymphoedema, ILCO (colorectal and ostomy cancer), Lung Cancer Association, and Gynsam–National Gynaecological Cancer Patients Coalition.

[6] 1.6 Million Club: a non-profit women's health organisation; at the time of organisational launch, there were 1.6 million women above 45 years of age in Sweden.

**Table 4. Top 10 payment categories and goals in Sweden (2014–18).**

| Category | Payment, € (%)[1] | Median, € (IQR) | n (%)[2] | Example (abbreviated) |
|---|---|---|---|---|
| Contributions to costs of events organised by recipients or third parties | 2 300 750 (35.7) | 2 170 (1 088 to 3 451) | 767 (57.4) | Contribution to cover rent, lecturers, study material, food, marketing of meetings, and media processing for event |
| Support and help | 1 325 363 (20.5) | 4 683 (1 984 to 10 266) | 104 (7.8) | Support for creation of website for disseminating information on breast cancer |
| Partnership arrangements | 1 021 968 (15.8) | 5 484 (2 564 to 11 020) | 104 (7.8) | A joint seminar to discuss a Hepatitis C national action plan with government representatives and politicians |
| Sponsorships | 831 430 (12.9) | 5 466 (2 149 to 10 587) | 108 (8.1) | Sponsorship of annual campaign on prostate cancer and men's health |
| Grants | 248 411 (3.9) | 5 601 (3 167 to 7 671) | 31 (2.3) | Grant to be used for leaflet on metastatic breast cancer, multiple events for young women with breast cancer, four advertisements in patient organisation magazine, and printed materials to support an app. |
| Form of funding unclear | 181 610 (2.8) | 1 024 (0 to 2 964) | 32 (2.4) | Un-specified project costs |
| More than one distinct payment form | 154 362 (2.4) | 6 721 (2 192 to 14 136) | 13 (1.0) | Four different payments to four different projects organised by the patient organisation |
| Travel, accommodation and registration fees | 67 566 (1.0) | 1 002 (560 to 2 062) | 46 (3.4) | Cover cost of travel, transfer, accommodation and meals for one patient organisation member attending conference |
| Sponsorship of participation at events organised by drug companies | 66 134 (1.0) | 548 (296 to 1 400) | 39 (2.9) | Cover cost of one speaker from patient organisation to attend company conference |
| Fees for service and consultancy (including travel and accommodation) | 54 548 (0.8) | 553 (339 to 1 108) | 34 (2.5) | Payment to patient organisation for 29 hours of work checking text for a company web site |
| **Goal** | **Payment, € (%)[1]** | **Median, € (IQR)** | **n (%)[2]** | **Example (abbreviated)** |
| Communication in general, media, meetings, online, publications, skills development | 2 041 674 (31.7) | 1 885 (983 to 3 361) | 609 (30.6) | Financial support for online initiative to acknowledge World Pancreatic Cancer Day |
| Advocacy, campaigning, and disease awareness | 1 707 507 (26.5) | 3 551 (2 136 to 8 177) | 242 (18.1) | Support for communication activities (media invitation and event) seeking to raise awareness of burden of kidney disease |
| Education and training | 736 816 (11.4) | 3 175 (1 264 to 6 017) | 146 (10.9) | Sponsorship to organise two-day educational meeting |
| More than one distinct purpose mentioned | 733 274 (11.4) | 3 620 (1 758 to 10 412) | 42 (3.1) | Contribution to maintenance, update and development of website, and to the work of certifying incontinence clinics in Sweden |
| Patient support | 356 801 (5.5) | 2 996 (1 509 to 5 609) | 51 (3.8) | Funding for evaluating and developing digital patient tool |
| Policy engagement | 204 416 (3.2) | 3 289 (3 144 to 5 186) | 43 (3.2) | Support for creation of a platform for stakeholders to meet, discuss and debate cancer issues, including with purpose to influence decision-makers to provide more resources to cancer care |
| Accessing or paying for organisation's expertise | 146 256 (2.3) | 548 (224 to 1 112) | 85 (6.4) | Financial compensation and travel expenses for patient organisation member to give lecture at internal company conference |
| Support for fundraising | 127 009 (2.0) | 6 048 (5 041 to 12 382) | 8 (0.6) | Financial support to organise charity-gathering event before annual World Diabetes Day |
| Inputting to organisation's work via membership, partnership, sponsorship or support | 108 399 (1.7) | 2 134 (941 to 3 456) | 35 (2.6) | Support on site when preparing and serving Christmas lunch during organisation's Christmas celebration |
| Funding for awards | 78 800 (1.2) | 5 292 (1 400 to 5 680) | 15 (1.1) | Company donation to a fund that annually award research grants on bleeding disorders |
| Research | 59 544 (0.9) | 5 041 (1 955 to 9 526) | 11 (0.8) | For patient organisation to help with a company's European study, with members encouraged to fill in survey linked from the organisation's Facebook page. |

[1] Percent of total value of payments (€6,449,224)

[2] Percent of total number of payments (1,337)

Furthermore–looking at the *goal* of the funding–public involvement, including "communication", "advocacy, campaigning and disease awareness" and "policy engagement" together attracted the lion's share of the funding (61.3%: €3,953,597). In contrast, "support for patients" attracted just 5.5% (€356,801) and "research-related activities" 0.9% (€59,544).

There were some payments for which the category or the goal of funding (or both) could not be determined due to lack of information, totalling €181,610 (2.8%) and €59,228 (0.9%), respectively.

## Conditions and diseases

Table 5 reports on broad condition areas and narrower disease areas by amount of funding received. Of the 17 ICD-10 condition areas, the top five amassed 69.6% (€4,485,252), with neoplasms alone attracting 37.5% (€2,419,186). This was followed by infectious and parasitic (9.1%; €590,026), endocrine, nutritional and metabolic (9.1%; €587,605), musculoskeletal (6.9%; €446,718) and digestive system (6.8%; €441,717) diseases.

However, within the broader condition areas, we observed a clear hierarchy among diseases. Thus, within neoplasms, cancers of breast, blood (lymphoid, hematopoietic and related tissues) and prostate amassed the bulk of funding (60.6%) whereas, for example, cancer of the female genital organs (1.9%) and skin (0.1%) received comparably little funding. This pattern was even more evident for some other condition areas. For example, virtually all funding for infectious and parasitic diseases went to HIV and hepatitis C. Overall, the top ten diseases in terms of funding accumulated 58.6% (€3,777,683). By far, most funding went to breast cancer with 10.3% (€663,774), followed by HIV (7.3%; €469,074) and blood cancers (7.2%; €463,378).

As shown in Table 5, companies tended to invest selectively and heavily in certain disease areas, which had a major effect on the overall pattern of funding. Pfizer, for example, provided most of the funding of breast cancer advocacy (74.1%; €492,074). Similarly, Astellas provided 43.9% (€148,674) of the funding of prostate cancer advocacy, whereas GlaxoSmithKline dominated HIV (76.9%; €360,702) and AbbVie Psoriasis (47.2%; €103,626) and hepatitis C (95.4%; €96,526) funding. At the same time, certain disease areas had many funders, such as cancers in general (n = 16) and blood cancers (n = 10) and diabetes (n = 9), resulting in overall substantial sums.

## The politician's week: Joint industry-patient organisation lobbying

A particular aspect of the industry funding landscape in Sweden was the substantial and recurrent funding of patient organisations during the yearly Politician's Week, or Almedalen Week. Payment reports shows that the Politician's Week is an important arena where patient organisations, together with one or more companies, seek to frame policy debates on certain diseases and treatments, especially cancers. Between 2014 and 2018, 8.7% (n = 116) of all payment reports pertained to the Politician's Week, with a total value of €704,763 (10.9%). Nineteen patient organisations (24.7%) received funding from 22 companies (46.8%). The companies mostly contributed to "events" organised by patient organisations (51.0%; €359,458), followed by "partnership arrangements" between patient organisations and industry (23.2%; €163,433). The goals of the Politician's Week payments were mainly "advocacy and disease awareness" attracting 46.6% (€327,938), followed by "policy engagement" (22.2%; €156,643) and "communication" (20.0%; €140,776). About half of the funding during the Politician's Week concerned neoplasms (51.1%; €360,186), mostly cancer in general (all through The Network Against Cancer) (56.8%; €204,694). Payments combining "policy engagement" and neoplasms attracted 21.3% (€150,455) of all funding during the Politician's Week, representing as much as 73.6% of the total funding for "policy engagement" between 2014–2018.

**Table 5. Reported drug industry funding of patient organisations in Sweden across condition and disease areas (2014–18).**

| Condition and Disease area | ICD-10 code | Payment, € (%) | n (%) | Supporting companies | Main donor,€ (%) |
|---|---|---|---|---|---|
| *Neoplasms* | *C00-D49* | *2 419 186 (37.5)[1]* | *606 (45.3)[2]* | *22* | *Pfizer, 619 662 (25.6)[3]* |
| Breast | C50 | 663 774 (27.4)[3] | 62 (10.2)[4] | 5 | Pfizer, 492 074 (74.1)[5] |
| Lymphoid, hematopoietic and related tissue | C81-C96 | 463 378 (19.2) | 121 (20.0) | 10 | Celgene, 121 363 (26.2) |
| General | C00-D49 | 416 250 (17.2) | 122 (20.1) | 16 | Roche, 52 592 (12.6) |
| Male genital organs (all prostate) | C61 | 338 684 (14.0) | 134 (22.1) | 8 | Astellas, 148 674 (43.9) |
| Respiratory and intrathoracic organs (all lung) | C30-C39 | 153 225 (6.3) | 50 (8.3) | 11 | Roche, 43 012 (28.1) |
| Digestive organs | C15-C26 | 129 852 (5.4) | 46 (7.6) | 7 | Roche, 60 015 (46.2) |
| Neuroendocrine | C7A | 86 227 (3.6) | 35 (5.8) | 3 | Novartis, 31 075 (36.0) |
| Of uncertain behavior, polycythemia vera and myelodysplastic syndromes | D37-D48 | 81 847 (3.4) | 8 (1.3) | 2 | Novartis, 70 718 (86.4) |
| Female genital organs | C51-C58 | 46 487 (1.9) | 15 (2.5) | 4 | Roche, 19 604 (42.2) |
| Urinary tract | C64-C65 | 19 122 (0.8) | 7 (1.2) | 4 | Pfizer, 8 705 (45.5) |
| Eye, brain and other parts of central nervous system | C71 | 19 165 (0.8) | 4 (0.7) | 1 | Roche, 19 165 (100) |
| Skin | C49 | 1 523 (0.1) | 2 (0.3) | 2 | Eli Lilly, 975 (64.0) |
| *Certain infectious and parasitic* | *A00-B99* | *590 026 (9.1)* | *105 (7.9)* | *9* | *GSK, 360 702 (62.1)* |
| HIV | B20-B24 | 469 074 (79.5) | 60 (57.1) | 5 | GSK, 360 702 (76.9) |
| Hepatitis C | B18.2 | 109 481 (18.6) | 39 (37.1) | 4 | AbbVie, 96 526 (95.4) |
| Both HIV and Hepatitis C | B18.2, B20 | 8 391 (1.4) | 4 (3.8) | 1 | AbbVie, 8 391 (100) |
| Other | A84.1 | 3 080 (0.5) | 2 (1.9) | 1 | Pfizer, 3 080 (100) |
| *Endocrine, nutritional and metabolic* | *E00-E90* | *587 605 (9.1)* | *121 (9.0)* | *14* | *Sanofi, 240 600 (40.9)* |
| Diabetes (Type 1 or 2) | E10, E11 | 340 972 (58.0) | 79 (65.3) | 9 | Boehringer Ingelheim, 86 889 (25.5) |
| Familial hypercholesterolemia | E78.0 | 162 742 (27.7) | 6 (5.0) | 2 | Sanofi, 157 701 (96.9) |
| Other | E20.9; E27.1; E34.3; E66; E75.22; E76; E85.1; E88.01 | 65 318 (11.1) | 24 (19.8) | 6 | Shire, 24 255 (37.1) |
| Fabry Disease | E75.21 | 18 086 (3.1) | 11 (9.1) | 2 | Shire, 9 125 (50.4) |
| *Musculoskeletal system and connective tissue* | *M00-99* | *446 718 (6.9)* | *78 (5.8)* | *13* | *124 141 (27.8) AbbVie* |
| Rheumatisms (various forms) | M05-06; M08 | 319 150 (71.4) | 45 (57.7) | 11 | AbbVie, 124 141 (38.9) |
| Osteoporosis | M80-M81 | 115 151 (25.8) | 27 (34.6) | 3 | Amgen, 99 854 (86.8) |
| Other | M32; M45 | 12 417 (2.8) | 6 (7.7) | 2 | Novartis, 6 575 (52.9) |
| *Digestive system* | *K00-K95* | *441 717 (6.8)* | *53 (4.0)* | *11* | *AbbVie, 141 150 (32.0)* |
| Noninfective enteritis and colitis including Crohn's | K50-52 | 302 435 (68.5) | 25 (47.2) | 5 | AbbVie, 138 017 (61.2) |
| Other | K00-95; K59; K90 | 139 282 (31.5) | 28 (52.8) | 9 | Takeda, 82 972 (59.6) |
| *Nervous system* | *G00-99* | *437 788 (6.8)* | *83 (6.2)* | *17* | *Biogen, 126 091 (28.9)* |
| Other | G10-G99, G12; G24; G40; G44 | 186 102 (42.5) | 34 (41.0) | 12 | Novartis, 40 241 (21.6) |
| Multiple Sclerosis | G35 | 143 060 (32.7) | 23 (27.7) | 6 | Biogen, 106 122 (74.2) |

(*Continued*)

**Table 5.** (*Continued*)

| Condition and Disease area | ICD-10 code | Payment, € (%) | n (%) | Supporting companies | Main donor,€ (%) |
|---|---|---|---|---|---|
| Parkinson's, Alzheimer's | G20; G30 | 108 626 (24.8) | 26 (31.3) | 8 | AbbVie, 77 334 (71.2) |
| *Blood and blood-forming organs and certain disorders involving the immune mechanism* | *D50-89* | *397 759 (6.2)* | *81 (6.1)* | *11* | *CSL Behring, 115 213 (29.0)* |
| Haemophilia | D65-D69 | 321 133 (80.7) | 64 (79.0) | 11 | Bayer, 90 404 (28.2) |
| Immunodeficiency | D84 | 50 800 (12.8) | 12 (14.8) | 4 | CSL Behring, 42 966 (84.6) |
| Other | D56; 61; 84.1 | 25 826 (6.5) | 5 (6.2) | 4 | Novartis, 22 865 (88.5) |
| *Circulatory system* | *I00-99* | *297 491 (4.6)* | *50 (3.7)* | *10* | *Pfizer, 94 870 (31.9)* |
| General | I10-I99; I48; I26-I52 | 187 310 (63.0) | 23 (46.0) | 8 | Pfizer, 81 825 (43.7) |
| Atrial fibrillation | I48 | 55 387 (18.6) | 10 (20.0) | 5 | Boehringer Ingelheim, 36 584 (66.1) |
| Cerebral infarction/Stroke | I63 | 30 789 (10.3) | 4 (8.0) | 1 | Bayer, 30 789 (100) |
| Others | I27; I81-I82; I89 | 26 884 (9.0) | 14 (28.0) | 4 | Actelion, 9 924 (36.9) |
| *Skin and subcutaneous tissue* | *L00-L99* | *245 964 (3.8)* | *36 (2.7)* | *9* | *AbbVie, 129 599 (52.7)* |
| Psoriasis | L40; L40.50 | 219 709 (89.3) | 32 (88.9) | 8 | AbbVie, 103 626 (47.2) |
| Other | L20; L73.2 | 26 255 (10.7) | 4 (11.1) | 2 | AbbVie, 25 973 (98.9) |
| *Various* | *N/A* | *200 863 (3.1)* | *36 (2.7)* | *14* | *Pfizer, 55 326 (27.5)* |
| Women's health | N/A | 108 758 (54.1) | 16 (44.4) | 5 | Pfizer, 55 326 (50.9) |
| Other | N/A | 92 105 (45.9) | 20 (55.6) | 11 | Sanofi, 33 842 (36.7) |
| *Respiratory system* | *J00-J99* | *180 235 (2.8)* | *37 (2.8)* | *8* | *Pfizer, 52 060 (28.9)* |
| COPD | J40-J44 | 148 958 (82.6) | 27 (73.0) | 6 | Pfizer, 52 060 (34.9) |
| Asthma | J45 | 17 985 (10.0) | 4 (10.8) | 2 | GlaxoSmithKline, 16 802 (93.4) |
| Lung fibrosis | J84.1 | 13 292 (7.4) | 6 (16.2) | 2 | Roche, 7 947 (59.8) |
| *Certain conditions originating in the perinatal period* | *P00-P96* | *67 177 (1.0)* | *16 (1.2)* | *3* | *AbbVie, 62 844 (93.5)* |
| Preterm newborn | P07 | 67 177 (100) | 16 (100) | 3 | AbbVie, 62 844 (93.5) |
| *Congenital malformations, deformations and chromosomal abnormalities* | *Q00-Q99* | *55 330 (0.9)* | *15 (1.1)* | *4* | *Novartis, 39 454 (71.3)* |
| Tuberous sclerosis | Q85.1 | 39 454 (71.3) | 9 (60.0) | 1 | Novartis, 39 454 (100) |
| Other | Q61, Q87.1, Q96 | 15 876 (28.7) | 6 (40.0) | 3 | Otsuka, 11 404 (71.8) |
| *Mental, Behavioural and Neurodevelopmental* | *F01-F99* | *42 911 (0.7)* | *13 (1.0)* | *6* | *Janssen, 17 192 (40.1)* |
| Schizophrenia | F20 | 29 250 (68.2) | 7 (53.8) | 4 | Janssen, 15 117 (51.7) |
| ADHD | F90 | 10 373 (24.2) | 5 (38.5) | 4 | Novartis, 5 716 (55.1) |
| Other | F11.10 | 3 288 (7.7) | 1 (7.7) | 1 | Shire, 3 288 (0.1) |
| *Symptoms, signs and abnormal clinical and laboratory findings, not elsewhere classified* | *R00-99* | *20 176 (0.3)* | *4 (0.3)* | *2* | *Astellas, 14 884 (73.8)* |
| Various | R32; R61 | 20 176 (100) | 4 (100) | 2 | Astellas, 14 884 (73.8) |
| *Eye and adnexa* | *H00-H59* | *10 959 (0.2)* | *1 (0.1* | *1* | *Santen, 10 959 (100)* |
| Glaucoma | H40 | 10 959 (100) | 1 (100) | 1 | Santen, 10 959 (100) |
| *Genitourinary system* | *N00-99* | *7 319 (0.1)* | *4 (0.3)* | *4* | *Otsuka, 2 879 (39.3)* |

(*Continued*)

**Table 5.** (Continued)

| Condition and Disease area | ICD-10 code | Payment, € (%) | n (%) | Supporting companies | Main donor,€ (%) |
|---|---|---|---|---|---|
| Various kidney | N25.81 | 7 319 (100) | 4 (100) | 4 | Otsuka, 2 879 (39.3) |

[1] Percent of total value of payments (€6,449,224)

[2] Percent of total number of payments (1,337)

[3] Percent of total value of payments to condition area

[4] Percent of total number of payments to condition area

[5] Percent of total value of payments to disease area

Notably, 68.1% (n = 79) of all reported Politician's Week payments pertained to activities co-funded by more than one company (52.8%; €372,154), often targeting policy- and decision-makers through "advocacy" (29.2%; €108,563) or "policy engagement" (27.6%; €102,882). Co-funded activities could involve as many as twelve companies, for example, aiming to lift the issue of regional inequalities in the use of cancer drugs onto the political agenda prior to the September 2018 parliamentary elections; or, as in 2015, a seminar on the "patient perspective" which, in addition to cancer patient organisation representatives, included speakers such as the Minister for Health and other high-level politicians, and the Chairpersons of the Swedish Medical Association and the Swedish Association of Health Professionals.

## Commercial motives for funding

Table 6 shows the concordance between commercialisation of drugs and the funding of patient organisations in the ten most funded disease areas (including cancer in general) for the top ten donors overall. If a company marketed at least one drug in a disease area, there was an 83% chance that it supported a patient organisation in the disease area (κ = 0.78, 95% confidence interval: 0.66–0.90). Companies only supported patient organisations in disease areas linked to their drug portfolio. AbbVie's support of Hemophilia patient advocacy appears at first to be an

**Table 6. Drug commercialisation and patient organisation funding in the ten most funded disease areas for the top ten drug industry donors in Sweden (2014–18).**

|  | Breast can. | | HIV | | Blood can. | | Gener. can. | | Diab. | | Prost. can. | | Hemo-ph. | | Rheu-ma. | | Enter., Col., Croh. | | Psor. | |
|---|---|---|---|---|---|---|---|---|---|---|---|---|---|---|---|---|---|---|---|---|
|  | F[1] | D[2] | F | D | F | D | F | D | F | D | F | D | F | D | F | D | F | D | F | D |
| Pfizer | Y | Y | N | N | Y | Y | Y | Y | N | N | N | N | Y | Y | Y | Y | N | Y | Y | Y |
| AbbVie | N | N | Y | Y | Y | Y | Y | Y | N | N | Y | Y | Y | N[3] | Y | Y | Y | Y | Y | Y |
| Sanofi | N | N | N | N | N | Y | Y | Y | Y | Y | Y | Y | N | N | N | Y | N | N | N | Y |
| Roche | Y | Y | N | N | Y | Y | Y | Y | N | N | N | N | Y | Y | Y | Y | N | N | N | N |
| Novartis | Y | Y | N | N | Y | Y | Y | Y | N | Y | N | N | Y | Y | Y | Y | N | N | Y | Y |
| GSK | N | N | Y | Y | N | N | N | N | N | N | N | N | N | N | N | N | N | N | N | Y[4] |
| Janssen | N | Y | Y | Y | Y | Y | Y | Y | Y | Y | Y | Y | N | N | Y | Y | Y | Y | Y | Y |
| Bayer | N | N | N | N | N | N | Y | Y | Y | Y | Y | Y | Y | Y | N | N | N | N | N | N |
| Celgene | N | Y | N | N | Y | Y | Y | Y | N | N | N | N | N | N | Y | Y | N | N | Y | Y |
| **Amgen** | Y | Y | N | N | Y | Y | Y | Y | N | N | Y | Y | N | Y | Y | Y | N | N | N | Y |

[1] Funding: yes/no

[2] Drug(s): yes/no

[3] See text for details

[4] GSK markets topicals to treat skin conditions: Betnovat (approved 1965), Dermovat (1976) and Flutivate (1993)

exception to this rule; however, hundreds of hemophilics in Sweden were infected by hepatitis C in the 1980s due to the use of virus-contaminated blood in transfusions, and AbbVie markets hepatitis C drugs.

To further investigate the possible link between commercial motives and funding, we assessed the correlation between the number of new drugs in 2014–2018 marketed by LIF companies in different condition areas and payments to patient organisations in those condition areas. Overall, 139 new drugs were marketed by LIF companies (n = 42). As many as one third (n = 46) were cancer drugs, followed by drugs for treating endocrine, nutritional and metabolic disorders (n = 23; 16.5%), and infectious and parasitic disorders (n = 19; 13.7%). Together, these three condition areas accounted for 56% of the funding over the study period (Table 5). In contrast, there were only three new drugs for treating mental, behavioural and neurodevelopmental disorders, and only one for treating ophthalmologic conditions. Across condition areas, there was a very strong, positive monotonic correlation between the number of new drugs marketed by LIF companies and the number of payments ($R_s$ = 0.85, n = 16, p = 0.00003) and the value of those payments ($R_s$ = 0.78, n = 16, p = 0.0004). There was also a very strong, positive monotonic correlation between the number of LIF companies marketing new drugs in different condition areas and the number of payments ($R_s$ = 0.84, n = 16, p = 0.00005) and the value of those payments ($R_s$ = 0.77, n = 16, p = 0.0005) in those condition areas. Results were similar for the subset of LIF companies (n = 30) that marketed new drugs (n = 118; 85%) and reported payments over the study period (€5,868,363; 90.0%): $R_s$ = 0.84–0.75, n = 16, p<0.0009, for all analyses.

## Discussion

This study's finding of close relations between patient organisations and pharmaceutical companies is consistent with what has been reported from other countries, including the UK [4, 26], United States [13, 14, 17, 28, 35], Australia [11, 27], Canada [12], Italy [36] and Finland [29]. Between 2014 and 2018, 77 Swedish patient organisations were reported to receive support from 46 companies, involving 1,337 payments with a cumulative value of roughly €6.4m. In addition, there were 75 (5.3%) payments reported which lacked information on the value of the payment, and which were therefore excluded from the analysis. The support was dominated by small payments, many of which were provided to cover costs of events organised by patient organisations, also consistent with findings from other countries [26, 27]. However, our data shows how even small payments may result in substantial financial support from industry since companies may co-fund projects, or one company may repeatedly fund the same or similar projects.

To contextualise our findings, and to enable broader generalisations, we tabulate selected features of industry payments in Sweden (this study) and the UK (reported in previous study using similar methodology) [26] (Table 7). Taking population size into account, the total value of payments was about 1.5 times greater in the UK, and the number of payments 1.9 times greater. There are two likely contributors to this difference. First, there were more reporting companies and recipient patient organisations in the UK. This may reflect country differences in, for example, drug commercialisation strategies (e.g., leading global role of UK drug market), government administration (e.g., healthcare in the UK is devolved to England, Scotland, Wales and Northern Ireland, possibly potentiating UK country-specific patient organisations) and availability of other funding (e.g., government funding seems more prominent in Sweden, possibly reducing the need for commercial sources of funding [33]). Second, large-size payments were more common in the UK, and they were often of a greater value, as evidenced by the larger median and upper quartile values. Notably, many large-size payments in the UK

**Table 7. Comparison of reported drug company payments to patient organisations in Sweden and the UK.**

| Selected variables | Sweden (2014–18) | UK (2012–16) |
|---|---|---|
| Population size | ~ 10m | ~ 65m |
| Value of payments, € | ~ 6.4m | ~ 65.1m |
| No. of payments | 1337 | 4572 |
| Median (IQR) | 2 411 (1 024 to 4 569) | 5 112 (686–11 984) |
| No. of companies | 46 | 64 |
| No. of patient organisations | 77 | 508 |
| Funding by top 10 companies | 68% | 69% |
| Public involvement | 61% | 31% |
| Support to patients | 6% | 6% |
| Research-related | 1% | 25% |
| Neoplasms | 38% | 36% |
| Endocrine, nutritional and metabolic | 9% | 11% |
| Infectious and parasitic | 9% | 8% |

were linked to research-related activities which accounted for 25% of the value of payments but only 3% of the number of payments. In Sweden, we found almost no funding of research-related activities. In contrast, most funding (61%) went to public involvement activities, encompassing communication, advocacy, campaigning, disease awareness and policy engagement, which typically involves smaller-size payments. This finding suggests marked country-differences in the role of patient organisations in the development of new medicines.

Nevertheless, some striking similarities existed in funding patterns in Sweden and the UK. The hierarchy of funding across condition areas was very similar, with the same top 3 condition areas receiving virtually identical shares of funding in the two countries, including 38% and 36% for neoplasms in Sweden and the UK, respectively. This finding suggests that strong commercial motives pattern companies' funding decisions, including an impetus to support cancer patient advocacy following the many cancer drug launches [35, 37]. However, our study further strengthens this argument. First, we show how companies tend to invest selectively and heavily in diseases linked to their drug portfolios, which shaped the overall pattern of funding across conditions and diseases. Second, we show a very strong correlation between, on the one hand, the number of new drugs and number of companies marketing new drugs in different condition areas and, on the other hand, the payments to patient organisations in different condition areas.

Another striking resemblance between Sweden and the UK was the concentration of funding across donors. In Sweden, only ten companies provided 68% of reported funding; in the UK this was 69%. The picture of a limited number of companies dominating relations is strengthened by findings from the UK showing high concentration of payments at the donor level to healthcare professionals (50% for the top 10 companies) [38] and organisations, such as hospitals and primary care centres, (59% for the top 10 companies) [39].

The broad picture, then, is of relatively few companies dominating relations by channelling their funding to patient organisations (and most likely other actors) in disease areas linked to their drug portfolios. As a consequence, much funding has been awarded to organisations for (some) cancer patients, such as those suffering from breast, blood or prostate cancers, but also diabetes and HIV. In stark contrast, limited funding goes to organisations for patients facing other major public health challenges, such as mental illness, but for which there have been few or no recent drug launches. Indeed, the top ten areas in terms of disease burden in Sweden in 2017 were ischemic heart disease (8.5% of total Disability-Adjusted Life Years), low back pain

(6.1%), stroke (4.2%), headache disorders (3.9%), diabetes (3.7%), Alzheimer's disease (3.6%), COPD (3.6%), falls (3.3%), depressive disorders (3.1%) and lung cancer (2.8%) [40, 41]. Yet, diseases of the circulatory system only amassed 4.6% of the value of payments compared to cancer's 37.5%, despite ischemic heart disease being the leading cause of death and disability in Sweden (although decreasing since 2007), whilst mental, behavioural and neurodevelopmental disorders only received 0.7% of all funding, and depressive disorders were not represented at all.

A concern with this commercially patterned funding is how it may create inequalities between patient organisations representing different diseases not only in terms of differential resources but also differential access to medical and political expertise and the media [11]. This risk is exemplified by the extensive joint campaigning and lobbying around certain diseases, especially cancers, during Sweden's annual Politician's Week. At this political, social and media event certain patient organisations, with support from companies, gain access to politicians and other policy- and decision-makers as well as the media. We found that 10.9% of all funding, and 14.9% of all cancer patient group funding, concentrated on the Politician's Week, and a substantial portion of this went to policy engagement. The Politician's Week payments also underscores the importance of taking into account the industry-level agenda for understanding funding patterns. Thus, as many as twelve companies could fund a single event targeting high-level policy and decision-makers.

In addition, we believe that the close link between funding and commercialisation are concerning in light of the uncertainty around the therapeutic value of many new drugs [42, 43, 44, 45, 46]. The problem of therapeutic uncertainty was exemplified by the EMA's recent withdrawal of the soft tissue cancer drug Lartruvo–which the EMA had approved in 2016 on the basis of preliminary data–after the company failed to demonstrate the drug's benefits in a post-authorisation confirmatory trial [47]. More generally, a recent study found that more than half of randomised controlled trials for cancer drugs in Europe approved in 2014–2016 had flaws likely to exaggerate treatment benefits [48]. Only one-quarter measured survival as a key outcome, and fewer than half reported on patients' quality of life. Ten of these new cancer drugs–Cometriq, Mekinist, Zydelig, Vargatef, Imbruvica, Kyprolis, Darzalex, Ibrance, Opdivo and Empliciti–were marketed by companies that supported a corresponding Swedish patient organisation and/or the Network Against Cancer. The support included several key cases of policy engagement. We cannot make judgements regarding the impact of industry funding on any particular patient organisation or activity. However, there is arguably a danger that patient organisations–who are unlikely to have access to all trial documentation for novel drugs, nor the capacity to undertake independent review of this material–may become direct or indirect champions of drugs that are putting health systems under great financial strain but whose therapeutic benefits remain uncertain.

Prior research has debated the advantages and shortcomings of industry self-regulation and government regulation for ensuring public disclosure of payments to patient organisations. For example, Kang et al [28] suggested implementing a legal disclosure requirement in the United States modelled on the Physician Payments Sunshine Act, rather than self-regulation, to ensure centralised payment reporting by companies because government regulation is better at compelling universal compliance. However, the Swedish example shows that a centralised database can in principle be achieved with self-regulation. The Swedish example also shows that national drug industry trade groups can, should they chose to do so, go beyond the rules set out by the industry at the European level that only requires of companies to disclose shorter descriptions of payments to patient organisations on their websites on an annual basis [25]. In April 2020, the Swedish pharmaceutical industry trade group updated its Code of Practice with some further innovations that could also be adopted by other countries [49].

Most importantly, companies in Sweden should now report payments to individual 'expert' patients and caregivers in the database in addition to the payments to patient organisations. Moreover, companies are no longer allowed to pay for patient organisations' costs for travel, accommodation and food at meetings and conferences, unless the patient organisation representatives are acting as consultants for the company. This latter rule extends the ban that already existed on paying for health professionals' costs for travel, accommodation and food at meetings and conferences [49].

It is tempting to speculate that the industry's more stringent rules and disclosure requirements in Sweden compared to, for example, the UK [22, 26] relate to Sweden's historically more transparent approach to government and business regulation, including in the pharmaceutical area [50]. However, we note that for the disclosure of payments to healthcare professionals and organisations it is the industry in the UK, but not in Sweden, that established a centralised database of payments, and that several European countries have opted for government regulation to guarantee public disclosure of payments to healthcare professionals [23]. This underscores the point that Sweden's self-regulatory system also has significant room for improvement [32]. Most pertinent to this study, disclosure databases need to be downloadable in an analysable format (e.g. CSV) to permit efficient and independent analysis. In addition, there is no reason why payment reports should be deleted after three years as is permitted under the current rules. Much can be also done to improve the standardisation and consistency of reporting, for example by linking payment descriptions to pre-established payment categories and goals; ensuring that costs are reported in the same way by all companies; and that information is always complete and represent actual costs (see below); and requiring that companies specify–in a standardised way–the disease or diseases in relation to which the payment is made. Finally, the fact that more than 5% of reports lacked clear information on the value of the payment demonstrates the need for improving compliance and quality control of the database.

## Limitations and strengths

One key limitation of our study is that we had to rely on the information provided by companies with no possibility of independently verifying the data. Furthermore, whilst most drug companies active in Sweden are trade group members or subscribe to the Code, and are therefore expected to report in the database, there are exceptions, e.g. Vertex and Gedeon Richter; therefore, our study likely underestimates the volume of payments. A further limitation is that many companies appear to enter information in the database after a formal arrangement between parties has been reached, but before the payment has been made, which means they report expected or maximum costs rather than the actual cost. This is a major limitation because we cannot know if there are differences between expected or maximum costs and the real costs. Another limitation is that each data entry was extracted and coded by one author, but our spot checks by a second author did not identify any discrepancies. Finally, we had to exclude 75 reports (5.3%) that lacked information on the value of the payment. Because of these limitations the values we present should be interpreted with some caution.

A further limitation pertains to our analysis of the association between drug commercialisation and industry funding across conditions and disease areas. Most importantly, we cannot exclude that some of the observed differences are caused by differences in patient organisations' willingness to accept commercial funding. Related to this, we cannot know what effect, if any, funding had on patient organisations. Future studies should therefore increasingly investigate the perspectives and motives of patient organisations and their representatives [11].

That said, the industry payment reports in Sweden have clear advantages compared to previously used data sources. Most importantly, Sweden's centralised and searchable database of

payments means greater transparency and greater certainty that all disclosures are included in the analysis. Furthermore, our impression is that the Swedish database contains more detailed descriptions of the payments compared to the reports published on company websites. This seems to translate into less ambiguity regarding the purpose of the funding. Indeed, compared to the UK, likely as a result of more detailed information, much fewer payments were coded as unclear with respect to the goal of funding (0.9% vs. 7.6%). The more detailed information also allowed us to characterise the extensive patient organisation-industry joint campaigning during a national lobbying event that merits follow up in future research.

Finally, a key strength of our study is the use of the same methodology as in the previous UK study [26] which allowed us to identify similarities and differences in the pattern of funding between the countries. Future research should collect and analyse comparable data from a larger sample of countries to improve the generalisation of results.

## Conclusion

This Swedish study reveals close and widespread ties between patient organisations and drug companies. The broad picture is of a relatively limited number of companies dominating relations by supporting patient organisations in disease areas linked to their drug portfolios. The Swedish disclosure initiative's online database has clear advantages for identifying and understanding the nature of industry funding, but the lack of downloadability and standardisation made the analysis burdensome. It is possible that the close ties between patient organisations and companies reflect coinciding interests. However, because funding is commercially motivated it will create inequalities between patient organisations in different disease areas even if there are coinciding interests between donors and recipients. Furthermore, the strong association between drug commercialisation and industry funding is arguably problematic in light of the uncertainty surrounding the therapeutic value of many new and typically expensive drugs.

## Supporting information

**S1 Table. Coding manual for payment categories and goals.**
(DOCX)

**S2 Table. Drug industry funders of Swedish patient organisations (2014–18).**
(DOCX)

**S3 Table. Top 10 drug industry payments to Swedish patient organisations (2014–18).**
(DOCX)

## Author Contributions

**Conceptualization:** Shai Mulinari, Andreas Vilhelmsson, Piotr Ozieranski.

**Data curation:** Andreas Vilhelmsson.

**Formal analysis:** Shai Mulinari, Andreas Vilhelmsson.

**Funding acquisition:** Shai Mulinari, Piotr Ozieranski.

**Investigation:** Shai Mulinari.

**Methodology:** Shai Mulinari, Piotr Ozieranski.

**Project administration:** Shai Mulinari.

**Supervision:** Shai Mulinari.

**Writing – original draft:** Shai Mulinari, Andreas Vilhelmsson, Emily Rickard, Piotr
    Ozieranski.

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
