## [Decision Letter · Decision Letter 0]

22 May 2020

PONE-D-20-13197

Five years of pharmaceutical industry funding of patient organisations in Sweden: cross-sectional study of companies, patient organisations and drugs

PLOS ONE

Dear Dr. Mulinari,

Thank you for submitting your manuscript to PLOS ONE. After careful consideration, we feel that it has merit but does not fully meet PLOS ONE’s publication criteria as it currently stands. Therefore, we invite you to submit a revised version of the manuscript that addresses the points raised during the review process.

In addition to the points raised by the reviewers there some additional comments that I have that the authors should address:

Although the requirements from the EFPIA may be the most comprehensive available that does not necessarily make them optimal and beyond calling for reports to be made available in a downloadable format the authors should suggest other possible changes that should be made to the information that EFPIA requires.Did the two authors extract the data independently and how were disagreements resolved?Page 8, line 17: When the authors say "target the disease" does that mean that the particular disease has to be an approved indication for the drug?What statistical software was used and what level of significance was used?Pages 23-24: Some of the material from page 23, line 11 to page 24, line 10 should be divided up between Methods and Results. The purpose of the Discussion is to contextualize the results not to present new results. (The material from page 23, line 15 to page 24, line 1 does belong in the Discussion.)Page 26, line 12: It should be "media" not "medial".

We look forward to receiving your revised manuscript.

Kind regards,

Joel Lexchin, MD

Academic Editor

PLOS ONE

Journal Requirements:

Reviewers' comments:

Reviewer's Responses to Questions

**Comments to the Author**

1. Is the manuscript technically sound, and do the data support the conclusions?

Reviewer #1: Yes

Reviewer #2: Yes

Reviewer #3: Yes

2. Has the statistical analysis been performed appropriately and rigorously? 

Reviewer #1: Yes

Reviewer #2: I Don't Know

Reviewer #3: Yes

3. Have the authors made all data underlying the findings in their manuscript fully available?

Reviewer #1: No

Reviewer #2: No

Reviewer #3: Yes

4. Is the manuscript presented in an intelligible fashion and written in standard English?

Reviewer #1: Yes

Reviewer #2: Yes

Reviewer #3: Yes

5. Review Comments to the Author

Reviewer #1: The authors provide a descriptive analysis of pharmaceutical company funding of patient groups in Sweden. They took great pains to extract data from an online industry website where payments to patient groups are disclosed. Having participated in creating a similar database in Australia, I can well appreciate the pains the authors took to extract these data and the value in having such a dataset now publicly available and downloadable. I would encourage the authors to create a DOI for their dataset and to find an institutional repository to make this available. Beyond academia, it has great value to journalists and also patient groups.

The authors well-situate the current study internationally and clearly articulate the contribution this analysis will make. This analysis, while it replicates one conducted in the UK and echoes similarly analyses in Australia and the US, provides important comparative data. It also provides fascinating local context with the descriptive analysis of “Politician’s Week.”

The most important contribution perhaps is the analysis of the relationship between a drug company’s product portfolio and patterns in how payments to patient groups are allocated.

Overall, the study is clearly reported and rigorously conducted. I make the following minor comments.

Abstract:

- Mention country in the abstract text

Introduction:

- Suggest removing “potential” as adjective for “conflict of interest” (line 7).

- The first paragraph could perhaps more explicitly describe the concerns related to patient group-industry relationships and the broad implications for society/policy/health. Perhaps the authors could cite a concrete example that illustrates the concern about relationships (e.g. supporting policy positions that favour drug companies).

Methods:

- What was the justification for the 5-year period? Is this the time frame for which data are available? In either case, it would be interesting to know when LIF began reporting and the impetus to begin reporting (though better to describe in the Introduction, perhaps).

- In the section, “Association between drug commercialisation and industry funding,” I did not understand the distinction (or the relationship?) between the analysis described in the first paragraph compared with the second. The second analysis described seems more robust, comprehensive and also clearer in terms of how the relationship between commercialization and funding of patient groups was assessed.

- I would appreciate more information as to how the categories for “Category” and “Goal” were developed. I realize that the authors relied on a scheme developed in a past analysis, but at minimum, it would be valuable to describe whether the authors relied on categories/goals developed by the industry reporters, or whether it was more of an inductive categorization the authors conducted. A supplementary file with the coding manual defining each of these categories would also be helpful.

Results

- I am curious about your choice of the word “collaboration” to describe the patient advocacy-industry relationships, for example, at the beginning of the “46 pharmaceutical companies reported 1,412 collaborations with the local, regional or national branches of 77 patient organisations.” Is this the authors’ word choice or the way it is described in the database?

- Sentence on page 10, lines 12-15 I think should be two sentences (period instead of comma after Table 2).

- For the sentence, “Pfizer was the major donor with 14.8% of the reported funding whereas AbbVie had the greatest number of payments (10.9%; n=146),” perhaps provide numerators/denominators to distinguish between the two measures

- In Table 3, could you clarify whether the % of funding measure for the columns “Value” and “Main donor” are percent of the donor’s total funding of patient groups or percent of total pharma funding received? For all Tables (3, 4 and 5), it would be helpful to have a row at the top with the “Totals” to provide context for the proportions (which I assume are column proportions?)

- Table 4 is informative and I particularly appreciate the illustrative examples. However, I found the information much more meaningful for the “Goal” categories. I would suggest removing the “Category” data from the table and consolidating most of the categories and either presenting them in a separate table or reporting in the narrative text. Specifically, I wonder whether there is a meaningful difference between “Sponsorships” and “Contributions to costs of event” (a form of sponsorship), “Support and help” (arguably the point of sponsorship) and “Grants.” All seem to be a lump sum provided by the company for the patient groups’ activities. Could these be grouped? I don’t really understand how “Partnership Arrangements” is defined. The remaining categories are self-evident and clearly distinct.

- The phrase, “ten most funding drug companies” (used throughout) might be more clearly rephrased as “top ten contributors” or something

Discussion

- The Discussion is thoughtful and comprehensive.

- For Table 7 (which is helpful), could you provide additional rows to show the value and number of payments per capita? (e.g. value/population size)

- I would add to the discussion of accessibility of public disclosure data that it should not only be downloadable, but in an analyzable format. For example, for a long time in Australia, PDFs could be downloaded, but make analysis impossible. All data should be provided in CSV format for download.

Reviewer #2: Summary

The researchers seek to add to a body of country-specific research on pharmaceutical industry funding of patient advocacy organizations by describing how this funding is distributed among patient organizations in Sweden and assessing whether the financial support that companies provide has a commercial agenda that could influence patient advocacy. A secondary purpose is to evaluate the industry’s centralized database of pharmaceutical industry donations, which is unique to Sweden. The results support the authors’ claim that investments in the patient community are significant and the patterns of funding indicate commercial agendas. While a demonstrated effect on advocacy is neither possible from the data nor claimed, the data analysis and discussion clearly illustrate a mismatch between funding patterns and the main objectives of national health policies: therapeutic need, therapeutic value and fair resource distribution.

The methodology is appropriate and follows that of a similar study in the UK that three of the four authors co-authored. This allowed for comparable units of analysis between the two and direct comparison of the two data sets in the discussion. This is a positive feature and begins an important investigation of industry strategies of product promotion that crosses borders.

The research is carefully conducted, the text is clearly written, and the data support the authors’ claims. I recommend publication with minor clarifications, corrections, and additions.

Issues to be addressed

Major

• Tables 1 through 5 provide summary data; the raw data is not provided (i.e., listing by year of individual donations from each company to each organization with the inflation and currency exchange rate noted for each year).

• Text-table discrepancy: In Table 5, second row on p 15, the number of supporting companies is 17; in the text, p 19, line 7 says “cancers in general” had 16 funders.

• Page 30, line 1: Reference 1., Democratizing Health, lists two co-editors; Hans Löfgren is also an editor (the first of three listed on the book cover).

Minor

- Pp 2-3, line 2:25, 3:1-2: This list of conditions is confusing. Use colons instead of commas to distinguish the three top conditions from the sub-conditions within them.

- Change “North America” to “United States” on page 5, line 1, referring to the spur in research on disclosure policies in different countries (Canada and Mexico do not have the same level of disclosure as the U.S. and reference #27, p 32, l 20-22 is to a U.S. study).

- In Table 3, the “1.6 Million Club” for women over the age of 46 attracted substantial funding from 8 companies, but the relevant condition(s) is/are not obvious. Do the supporting companies and/or the events supported suggest what motivated these donations? How were donations to this group, and any others with ambiguous health constituency handled in the coding?

- In tables 2, 3, 4, 5 and S1, clearly indicate the meaning of column heading “n(%)”. I believe “n” refers to “Number of Reports” but had to comb the text to determine this.

- Check the manuscript for words that run together, e.g., cover page abstract: “betweennew drugs”, “Theassociation between” and “commercialisationand industry”

Questions and comments for the authors

1) The top payment category is “contributions to cost of events organized by recipients or third parties.” Do the authors know who these third parties are? Pharmaceutical companies often work in concert with public relations firms in their collaborations with groups (see ref 15, 187-188). Essentially the “third party” PR firm works for the company and shapes the event to further the marketing goals of the company; simply referring to a “third party” suggests a third independent actor.

2) A paragraph discussing the role of industry Codes of Practice and the relationship between voluntary and legally mandated disclosures would be helpful. In particular, the reader may question why the database in Sweden is centralized when other countries under EPFIA use decentralized databases. What are the avenues for achieving centralized (and downloadable) databases? How important are legal interventions to opening up transparency? Also, although the EPFIA and Swedish codes cited in the article are appropriate to the time period studied, both codes have been updated (EPFIA revised 2019; IFPMA update to take effect May 2020). A note on any changes to the codes would underline the fact that these codes are continually reviewed and would be valuable to researchers undertaking future investigations, especially if changes affect transparency in patient group relations. Finally, the Ethical Rules for the Pharmaceutical Industry in Sweden (2015: 57) states that, “Short versions of all contracts and agreements should also be available in the LIF Cooperation Database when activity is ongoing and at least for one month after the project is concluded.” Were any of these contracts viewed? If so, are they useful for assessing the industry’s use of the activities for commercial advantage?

3) The UK-Sweden comparison and the finding that substantial funding is devoted to one advocacy event could be listed in the “strengths” section. Research conducted in a country where English is not the dominant language or culture is an especially useful addition to the existing body of funding disclosure analyses.

4) Similarly, the focus on funding the national lobbying event, “Politicians’ Week” Is a novel finding and a strength that merits follow up in future research (e.g., federal governments in Canada and the US both have “Lobby Days” or “Advocacy Days”, which attract patient advocacy groups with industry funding).

Re: 2. Rigour of statistical analysis: In my assessment the use Cohen’s k and Spearman’s rank order correlation [page 9] are appropriate and support the authors' conclusions; however, I am a qualitative researcher and have limited familiarity with the choice and use of statistical tests for measuring associations.

Reviewer #3: This paper makes an important contribution to understand the recent history of patient organisation funding by the pharmaceutical industry in Sweden. It uses the appropriate level of nuance and analytical tools to describe the results and assess the association between companies' commercial interests and patient group funding, with a particular focus on recently marketed medicines. Moreover, it also confirms a number of previously observed trends with regards to industry funding of patient organisations in the UK.

One minor suggestion is to clarify on page 4, line 18 that EFPIA represents research-based pharmaceutical companies. The same with LIF (page 5, line 20). Could the authors clarify whether this study is likely to capture most of the companies supporting patient groups in Sweden, or whether there are important sub-populations of companies (i.e. generics?) that are not described here. Generic companies may, but are unlikely to, also support patient organisations. For example, generic industry associations support other groups such as universities through funded research chairs. Thank you.

6. PLOS authors have the option to publish the peer review history of their article (what does this mean?). If published, this will include your full peer review and any attached files.

Reviewer #1: Yes: Quinn Grundy, Assistant Professor, University of Toronto

Reviewer #2: Yes: Sharon Batt, PhD

Reviewer #3: Yes: Katrina Perehudoff

---

## [Author Response · Author response to Decision Letter 0]

4 Jun 2020

Comments from Editor

In addition to the points raised by the reviewers there some additional comments that I have that the authors should address:

1 Although the requirements from the EFPIA may be the most comprehensive available that does not necessarily make them optimal and beyond calling for reports to be made available in a downloadable format the authors should suggest other possible changes that should be made to the information that EFPIA requires.

Response: Thank you. We very much agree with this comment, and we have added two new paragraphs in the Discussion to address it. Please refer to our response to Reviewer 2. 

2 Did the two authors extract the data independently and how were disagreements resolved?

Response: We did not extract the data independently but divided the data extraction between us. However, when AV (second author) collected the data from 2017-2018 in April and May 2019 he checked the data collected by SM (first author) from June 2016 to December 2016 and found no discrepancies. We have added this to the Methods. We have also added that the lack of confirmation for all data is a Limitation (in the Limitation section). 

3 Page 8, line 17: When the authors say "target the disease" does that mean that the particular disease has to be an approved indication for the drug?

Response: Thank you for this question. Yes, our definition was based on the approved indication (i.e. it did not include off-label uses) but, at the same time, it was technically broader than the approved indication which can be quite narrow. For example, idelalisib (Zydelig) is approved in a subset of patients with CLL or follicular lymphoma, but we coded it as a blood cancer drug. To clarify that we based our judgment on the approved indication, we have reworded as follows (changes in bold):

“We used information on approved indications to determine whether or not companies marketed at least one drug in each of the ten selected diseases. For a drug to be considered marketed in a disease it had be indicated for the disease, e.g. drugs approved in HIV/AIDS patients for combating non-HIV infections, such as fungal infections, were not HIV drug. Similarly, analgesics indicated for cancer patients were not cancer drugs.” 

4 What statistical software was used and what level of significance was used?

Response: We have added ‘A p-value below 0.05 was considered significant. Prism 8.2.1 for Macintosh (GraphPad Software Inc.) was used all statistical analyses.’

5 Pages 23-24: Some of the material from page 23, line 11 to page 24, line 10 should be divided up between Methods and Results. The purpose of the Discussion is to contextualize the results not to present new results. (The material from page 23, line 15 to page 24, line 1 does belong in the Discussion.)

Response: We understand the point raised. However, we feel we are not presenting any new data here; we are simply summarising some key points. Importantly, all the data in Table 7 can be found in Tables 1-5 and in our UK study. For this reason, we would prefer keeping the Discussion as is, unless the Editor advises strongly against it. 

6 Page 26, line 12: It should be "media" not "medial".

Response: Thank you

Journal Requirements:

Response: The complete dataset will be available here: https://zenodo.org/record/3875140#.Xtit3S2HJm9

DOI: 10.5281/zenodo.3875140

Reviewers' comments:

Reviewer #1: The authors provide a descriptive analysis of pharmaceutical company funding of patient groups in Sweden. They took great pains to extract data from an online industry website where payments to patient groups are disclosed. Having participated in creating a similar database in Australia, I can well appreciate the pains the authors took to extract these data and the value in having such a dataset now publicly available and downloadable. I would encourage the authors to create a DOI for their dataset and to find an institutional repository to make this available. Beyond academia, it has great value to journalists and also patient groups.

The authors well-situate the current study internationally and clearly articulate the contribution this analysis will make. This analysis, while it replicates one conducted in the UK and echoes similarly analyses in Australia and the US, provides important comparative data. It also provides fascinating local context with the descriptive analysis of “Politician’s Week.”

The most important contribution perhaps is the analysis of the relationship between a drug company’s product portfolio and patterns in how payments to patient groups are allocated.

Overall, the study is clearly reported and rigorously conducted. I make the following minor comments.

Response: Thank you very much for the positive assessment of our study. The complete dataset will be available here: https://zenodo.org/record/3875140#.Xtit3S2HJm9

DOI: 10.5281/zenodo.3875140

Abstract:

- Mention country in the abstract text

Response: This has been added. 

Introduction:

- Suggest removing “potential” as adjective for “conflict of interest” (line 7).

Response: We have removed ‘potential’

- The first paragraph could perhaps more explicitly describe the concerns related to patient group-industry relationships and the broad implications for society/policy/health. Perhaps the authors could cite a concrete example that illustrates the concern about relationships (e.g. supporting policy positions that favour drug companies).

Response: Thank you for this suggestion. We have added an example of this: 

“For example, several industry-funded patient groups at the European Union level supported the industry’s efforts to relax the ban on direct-to-consumer advertising, but this agenda was strongly opposed by patient groups that did not accept industry funding [21].”

Methods:

- What was the justification for the 5-year period? Is this the time frame for which data are available? In either case, it would be interesting to know when LIF began reporting and the impetus to begin reporting (though better to describe in the Introduction, perhaps).

Response: The 5-year period was selected for two main reasons. One reason was data availability. As we note, industry rules allow reports to be deleted after three years, which sets a limit on the time frame for which data was available. Second, we felt that 3 years would not be enough to capture the association between commercialisation and funding, but we expected that 5 years could be reasonable and manageable. Therefore, we collected data at two points in time to ensure we had 5 years of data. We have added a sentence explaining this:

“We selected a 5-year period in order to have a fairly large but still manageable sample, but also because industry rules allow reports to be deleted after three years which limits data availability”

LIF has had the database in place since 2005. We have added this information to the introduction. 

Regarding the impetus to begin reporting, we believe that a statement about this would require a separate study. LIF says that the impetus was its longstanding commitment to transparency, clarity and accountability, but it’s possible that the main impetus was protecting the industry and self-regulation from reputational threats. Indeed, when LIF launched the database in 2005, they referred to the recently published report by UK House of Commons Health Committee on the Influence of the Pharmaceutical Industry, and acknowledged that some of the issues raised in the report were relevant to Sweden too. 

- In the section, “Association between drug commercialisation and industry funding,” I did not understand the distinction (or the relationship?) between the analysis described in the first paragraph compared with the second. The second analysis described seems more robust, comprehensive and also clearer in terms of how the relationship between commercialization and funding of patient groups was assessed.

Response: Thank you for this comment. We believe the analyses provide different pieces of information that together support the claim of a strong link between drug commercialisation and payments to patient organisations.

The first analysis is about the association between commercialisation of drugs by specific companies, for example Pfizer, and the funding of patient organisations in specific diseases, for example breast cancer. This analysis is important because it shows that companies are funding patient organisations in disease areas linked to their specific drug portfolio. Indeed, the results show that we can quite accurately predict if a drug company will support a patient organisation or not by looking at its drug portfolio in the ten selected diseases, so this provides a quite strong argument for the ‘commercialisation thesis’. 

The second analysis is about the association between commercialisation of new drugs and payments to patient organisations in broader condition areas. It is important because it shows the overall relationship between the number of new drugs/companies marketing drugs and payments. 

In the Discussion we make the following two statements: 

1. ‘companies tend to invest selectively and heavily in diseases linked to their drug portfolios, which shaped the overall pattern of funding across conditions and diseases.’ (Analysis 1)

2 there is a ‘very strong correlation between, on the one hand, the number of new drugs and number of companies marketing new drugs in different condition areas and, on the other hand, the payments to patient organisations in different condition areas’. (Analysis 2) 

- I would appreciate more information as to how the categories for “Category” and “Goal” were developed. I realize that the authors relied on a scheme developed in a past analysis, but at minimum, it would be valuable to describe whether the authors relied on categories/goals developed by the industry reporters, or whether it was more of an inductive categorization the authors conducted. A supplementary file with the coding manual defining each of these categories would also be helpful.

Response: This is an important comment. We have added the following explanation in the Methods:

“In short, the payment category codes were initially devised based on the codes used by EFPIA to categorise pharmaceutical company payments to healthcare organisations (e.g. hospitals, universities, medical associations), specifically, “grants”, “contributions to costs of events”, “travel, accommodation and registration fees”, “fees for service and consultancy”, and “sponsorship”’. These codes were then supplemented with an inductive approach for any emerging payment categories that were unique to the patient organisation payment descriptions, such as “support and help” [26]. Separately, payment goal was coded based on close iterative reading of payment descriptions and aggregating similar descriptions under the same codes [26]. When coding payment goals we looked for the main purpose of activities funded by drug companies.”

In addition, we have added a coding manual (S1 Table) defining each of the categories and goals, as suggested by the reviewer. This manual was developed in our previous UK study. 

Results

- I am curious about your choice of the word “collaboration” to describe the patient advocacy-industry relationships, for example, at the beginning of the “46 pharmaceutical companies reported 1,412 collaborations with the local, regional or national branches of 77 patient organisations.” Is this the authors’ word choice or the way it is described in the database?

Response: The official name of the database is the Collaboration Database. This is indicated in the Abstract and Methods where we use “collaboration database” with quotation marks. However, in light of the reviewer’s thoughtful comment, we have replaced ‘collaborations’ with ‘relationships’ in the quoted paragraph. 

- Sentence on page 10, lines 12-15 I think should be two sentences (period instead of comma after Table 2).

Response: Thank you. 

- For the sentence, “Pfizer was the major donor with 14.8% of the reported funding whereas AbbVie had the greatest number of payments (10.9%; n=146),” perhaps provide numerators/denominators to distinguish between the two measures

Response: Thank you. We have added the numerator: “Pfizer was the major donor with 14.8% (€954,234) …” The denominators are in the previous paragraph and in Table 1. 

- In Table 3, could you clarify whether the % of funding measure for the columns “Value” and “Main donor” are percent of the donor’s total funding of patient groups or percent of total pharma funding received? 

Response: It’s the % of total industry funding. We have clarified this in the table by adding footnotes. 

For all Tables (3, 4 and 5), it would be helpful to have a row at the top with the “Totals” to provide context for the proportions (which I assume are column proportions?)

Response: We have clarified this in the tables by adding footnotes.

- Table 4 is informative and I particularly appreciate the illustrative examples. However, I found the information much more meaningful for the “Goal” categories. I would suggest removing the “Category” data from the table and consolidating most of the categories and either presenting them in a separate table or reporting in the narrative text. Specifically, I wonder whether there is a meaningful difference between “Sponsorships” and “Contributions to costs of event” (a form of sponsorship), “Support and help” (arguably the point of sponsorship) and “Grants.” All seem to be a lump sum provided by the company for the patient groups’ activities. Could these be grouped? I don’t really understand how “Partnership Arrangements” is defined. The remaining categories are self-evident and clearly distinct.

Response: We understand the reviewer’s concern. However, we have now clearly defined each ‘Category’ code in S1 Table and also give more detail in Methods (see previous response). 

- The phrase, “ten most funding drug companies” (used throughout) might be more clearly rephrased as “top ten contributors” or something

Response: Thank you very much for this suggestion. We have changed the text accordingly.

Discussion

- The Discussion is thoughtful and comprehensive.

Response: Thank you very much.

- For Table 7 (which is helpful), could you provide additional rows to show the value and number of payments per capita? (e.g. value/population size)

Response: The table contains the population size of each country. For Sweden it’s about 10m, which makes it very easy to calculate all values/population. For the UK, it’s 65m, which is also makes it easy to calculate, especially since the value of payments in the UK is about €65m.

- I would add to the discussion of accessibility of public disclosure data that it should not only be downloadable, but in an analyzable format. For example, for a long time in Australia, PDFs could be downloaded, but make analysis impossible. All data should be provided in CSV format for download.

Response: Thank you for this very valuable comment. We have added that ‘databases should be downloadable in an analysable format (e.g. CSV) to permit efficient and independent analysis.’

Reviewer #2: Summary

The researchers seek to add to a body of country-specific research on pharmaceutical industry funding of patient advocacy organizations by describing how this funding is distributed among patient organizations in Sweden and assessing whether the financial support that companies provide has a commercial agenda that could influence patient advocacy. A secondary purpose is to evaluate the industry’s centralized database of pharmaceutical industry donations, which is unique to Sweden. The results support the authors’ claim that investments in the patient community are significant and the patterns of funding indicate commercial agendas. While a demonstrated effect on advocacy is neither possible from the data nor claimed, the data analysis and discussion clearly illustrate a mismatch between funding patterns and the main objectives of national health policies: therapeutic need, therapeutic value and fair resource distribution.

The methodology is appropriate and follows that of a similar study in the UK that three of the four authors co-authored. This allowed for comparable units of analysis between the two and direct comparison of the two data sets in the discussion. This is a positive feature and begins an important investigation of industry strategies of product promotion that crosses borders.

The research is carefully conducted, the text is clearly written, and the data support the authors’ claims. I recommend publication with minor clarifications, corrections, and additions.

Response: Thank you very much for the positive assessment of our study. 

Issues to be addressed

Major

• Tables 1 through 5 provide summary data; the raw data is not provided (i.e., listing by year of individual donations from each company to each organization with the inflation and currency exchange rate noted for each year)

Response: Thank you. The complete dataset will be available here: https://zenodo.org/record/3875140#.Xtit3S2HJm9

DOI: 10.5281/zenodo.3875140

• Text-table discrepancy: In Table 5, second row on p 15, the number of supporting companies is 17; in the text, p 19, line 7 says “cancers in general” had 16 funders.

Response: Thank you so much for spotting this mistake. The correct number is 16.

• Page 30, line 1: Reference 1., Democratizing Health, lists two co-editors; Hans Löfgren is also an editor (the first of three listed on the book cover).

Response: Thank you so much for spotting this mistake.

Minor

- Pp 2-3, line 2:25, 3:1-2: This list of conditions is confusing. Use colons instead of commas to distinguish the three top conditions from the sub-conditions within them.

Response: Thank you for this suggestion. 

- Change “North America” to “United States” on page 5, line 1, referring to the spur in research on disclosure policies in different countries (Canada and Mexico do not have the same level of disclosure as the U.S. and reference #27, p 32, l 20-22 is to a U.S. study).

Response: Thank you for this suggestion. 

- In Table 3, the “1.6 Million Club” for women over the age of 46 attracted substantial funding from 8 companies, but the relevant condition(s) is/are not obvious. Do the supporting companies and/or the events supported suggest what motivated these donations? How were donations to this group, and any others with ambiguous health constituency handled in the coding?

Response: Thank you for this question. In many cases we could identify the disease based on the project and payment descriptions. In the case of the 1.6 Million Club, 5 of 32 payments went to osteoporosis disease awareness, 2 payments were linked to cardiovascular disease and 2 to stroke. However, half of the payments to the 1.6 Million Club, worth €108,758, could not be differentiated at the level of a disease, and they were coded as ‘Women’s health’ which is part of the ‘Various’ condition category. Importantly, these are all reported in Table 5 under ‘Women’s health’. 

- In tables 2, 3, 4, 5 and S1, clearly indicate the meaning of column heading “n(%)”. I believe “n” refers to “Number of Reports” but had to comb the text to determine this.

Response: Thank you. We have clarified this in the tables by adding footnotes.

- Check the manuscript for words that run together, e.g., cover page abstract: “betweennew drugs”, “Theassociation between” and “commercialisationand industry”

Response: Thank you. We have fixed this in the cover page abstract. 

Questions and comments for the authors

1) The top payment category is “contributions to cost of events organized by recipients or third parties.” Do the authors know who these third parties are? Pharmaceutical companies often work in concert with public relations firms in their collaborations with groups (see ref 15, 187-188). Essentially the “third party” PR firm works for the company and shapes the event to further the marketing goals of the company; simply referring to a “third party” suggests a third independent actor.

Response: In the Swedish dataset we only found one clear example of using a PR firm. Roche paid a PR firm approx. EUR 500 to organise a prize ceremony event for the colorectal cancer prize which is handed out by the patient organisation every year. In addition, there were a few instances of paying for co-arrangements with the leading medical news outlet in Sweden, ‘Dagens Medicin’ [Medicine Today], including at the Politician’s week.

2) A paragraph discussing the role of industry Codes of Practice and the relationship between voluntary and legally mandated disclosures would be helpful. In particular, the reader may question why the database in Sweden is centralized when other countries under EPFIA use decentralized databases. What are the avenues for achieving centralized (and downloadable) databases? How important are legal interventions to opening up transparency? Also, although the EPFIA and Swedish codes cited in the article are appropriate to the time period studied, both codes have been updated (EPFIA revised 2019; IFPMA update to take effect May 2020). A note on any changes to the codes would underline the fact that these codes are continually reviewed and would be valuable to researchers undertaking future investigations, especially if changes affect transparency in patient group relations. Finally, the Ethical Rules for the Pharmaceutical Industry in Sweden (2015: 57) states that, “Short versions of all contracts and agreements should also be available in the LIF Cooperation Database when activity is ongoing and at least for one month after the project is concluded.” Were any of these contracts viewed? If so, are they useful for assessing the industry’s use of the activities for commercial advantage?

Response: Thank you for these important comments. 

Regarding the last point, we did look at some contracts, but we didn’t find them especially informative in terms of understanding the activity better. They were simply more “formal” versions of what was already in the summary database. 

We have added the following paragraph to address the reviewer’s other comments: 

‘Prior research has debated the advantages and shortcomings of industry self-regulation and government regulation for ensuring public disclosure of payments to patient organisations. For example, Kang et al [28] suggested implementing a legal disclosure requirement in the United States modelled on the Physician Payments Sunshine Act, rather than self-regulation, to ensure centralised payment reporting by companies because government regulation is better at compelling universal compliance. However, the Swedish example shows that a centralised database can in principle be achieved with self-regulation. The Swedish example also shows that national drug industry trade groups can, should they chose to do so, go beyond the rules set out by the industry at the European level that requires of companies to disclose shorter descriptions of payments to patient organisations on their websites on an annual basis [25]. In April 2020, the Swedish pharmaceutical industry trade group updated its Code of Practice with some further innovations that could also be adopted by other countries [49]. Most importantly, companies in Sweden should now report payments to individual ‘expert’ patients and caregivers in the database in addition to the payments to patient organisations. Moreover, companies are no longer allowed to pay for patient organisations’ costs for travel, accommodation and food at meetings and conferences, unless the patient organisation representatives are acting as consultants for the company. This latter rule extends the ban that already existed on paying for health professionals’ costs for travel, accommodation and food at meetings and conferences [49].

It is tempting to speculate that the industry’s more stringent rules and disclosure requirements in Sweden compared to, for example, the UK [22, 26] relate to Sweden’s historically more transparent approach to government and business regulation, including in the pharmaceutical area [50]. However, we note that for the disclosure of payments to healthcare professionals and organisations it is the industry in the UK, but not in Sweden, that established a centralised database of payments, and that several European countries have opted for government regulation to guarantee public disclosure of payments to healthcare professionals [23]. This underscores the point that Sweden’s self-regulatory system also has significant room for improvement [32]. Most pertinent to this study, disclosure databases need to be downloadable in an analysable format (e.g. CSV) to permit efficient and independent analysis. In addition, there is no reason why payment reports should be deleted after three years as is permitted under the current rules. Much can be also done to improve the standardisation and consistency of reporting, for example by linking payment descriptions to pre-established payment categories and goals; ensuring that costs are reported in the same way by all companies; and that information is always complete and represent actual costs (see below); and requiring that companies specify – in a standardised way – the disease or diseases in relation to which the payment is made. Finally, the fact that more than 5% of reports lacked clear information on the value of the payment demonstrates the need for improving compliance and quality control of the database.’ 

3) The UK-Sweden comparison and the finding that substantial funding is devoted to one advocacy event could be listed in the “strengths” section. Research conducted in a country where English is not the dominant language or culture is an especially useful addition to the existing body of funding disclosure analyses.

Response: Thank you for this important suggestion. We have added the following paragraph in the ‘Limitation and strengths’ section: 

‘Finally, a key strength of our study is the use of the same methodology as in the previous UK study [26] which allowed us to identify similarities and differences in pattern of funding between the countries. Future research should collect and analyse comparable data from a larger sample of countries to improve the generalisation of results.’

4) Similarly, the focus on funding the national lobbying event, “Politicians’ Week” Is a novel finding and a strength that merits follow up in future research (e.g., federal governments in Canada and the US both have “Lobby Days” or “Advocacy Days”, which attract patient advocacy groups with industry funding).

Response: Thank you for this important suggestion. We have added the following paragraph in the ‘Limitation and strengths’ section: ‘The more detailed information also allowed us to characterise the extensive patient organisation-industry joint campaigning during a national lobbying event that merits follow up in future research’.

Re: 2. Rigour of statistical analysis: In my assessment the use Cohen’s k and Spearman’s rank order correlation [page 9] are appropriate and support the authors' conclusions; however, I am a qualitative researcher and have limited familiarity with the choice and use of statistical tests for measuring associations.

Reviewer #3: This paper makes an important contribution to understand the recent history of patient organisation funding by the pharmaceutical industry in Sweden. It uses the appropriate level of nuance and analytical tools to describe the results and assess the association between companies' commercial interests and patient group funding, with a particular focus on recently marketed medicines. Moreover, it also confirms a number of previously observed trends with regards to industry funding of patient organisations in the UK.

Response: Thank you very much for the positive assessment of our study. 

One minor suggestion is to clarify on page 4, line 18 that EFPIA represents research-based pharmaceutical companies. The same with LIF (page 5, line 20). Could the authors clarify whether this study is likely to capture most of the companies supporting patient groups in Sweden, or whether there are important sub-populations of companies (i.e. generics?) that are not described here. Generic companies may, but are unlikely to, also support patient organisations. For example, generic industry associations support other groups such as universities through funded research chairs. Thank you.

Response: Thank you for this important comment. 

The LIFs rules have been ratified by the trade group for smaller life companies and the trade group for generic pharmaceuticals and biosimilars manufacturers. However, we found no reported payments from non-LIF companies including generics companies. 

In the Limitations section we make the following statement that addresses the important issue raised by the reviewer: ‘Furthermore, whilst most drug companies active in Sweden are trade group members or subscribe to the Code, and are therefore expected to report in the database, there are exceptions, e.g. Vertex and Gedeon Richter; therefore, our study likely underestimates the volume of payments.’

---

## [Editor Report · Decision Letter 1]

8 Jun 2020

Five years of pharmaceutical industry funding of patient organisations in Sweden: cross-sectional study of companies, patient organisations and drugs

PONE-D-20-13197R1

Dear Dr. Mulinari,

We’re pleased to inform you that your manuscript has been judged scientifically suitable for publication and will be formally accepted for publication once it meets all outstanding technical requirements.

However, there are a number of minor copy editing changes that need to be made:

Page 3, line 5: The wording of this sentence can lead to misinterpretation and I'd suggest starting the sentence with "A relatively few..."Page 4, line 23: Delete "of".Page 5, line 4: Spell out UK the first time the abbreviation is used.Page 9, line 8: It should be "drugs" not "drug".Page 10, line 14: Delete the period after "449" and replace it with a comma.

Kind regards,

Joel Lexchin, MD

Academic Editor

PLOS ONE
---

## [Editor Report · Acceptance letter]

10 Jun 2020

PONE-D-20-13197R1 

Five years of pharmaceutical industry funding of patient organisations in Sweden: cross-sectional study of companies, patient organisations and drugs 

Dear Dr. Mulinari:

I'm pleased to inform you that your manuscript has been deemed suitable for publication in PLOS ONE. Congratulations! Your manuscript is now with our production department. 

Kind regards, 

on behalf of

Prof. Joel Lexchin 

Academic Editor

PLOS ONE